# An in-plane magnetic chiral dichroism approach for measurement of intrinsic magnetic signals using transmitted electrons

Dongsheng Song[1], Amir H. Tavabi[2], Zi-An Li[2], András Kovács[2], Ján Rusz[3], Wenting Huang[4,5], Gunther Richter[5], Rafal E. Dunin-Borkowski[2] & Jing Zhu[1]

Electron energy-loss magnetic chiral dichroism is a powerful technique that allows the local magnetic properties of materials to be measured quantitatively with close-to-atomic spatial resolution and element specificity in the transmission electron microscope. Until now, the technique has been restricted to measurements of the magnetic circular dichroism signal in the electron beam direction. However, the intrinsic magnetization directions of thin samples are often oriented in the specimen plane, especially when they are examined in magnetic-field-free conditions in the transmission electron microscope. Here, we introduce an approach that allows in-plane magnetic signals to be measured using electron magnetic chiral dichroism by selecting a specific diffraction geometry. We compare experimental results recorded from a cobalt nanoplate with simulations to demonstrate that an electron magnetic chiral dichroism signal originating from in-plane magnetization can be detected successfully.

[1] National Center for Electron Microscopy in Beijing, Key Laboratory of Advanced Materials (MOE) and The State Key Laboratory of New Ceramics and Fine Processing, School of Materials Science and Engineering, Tsinghua University, Beijing 100084, China. [2] Ernst Ruska-Centre for Microscopy and Spectroscopy with Electrons and Peter Grünberg Institute, Forschungszentrum Jülich, D-52425 Jülich, Germany. [3] Department of Physics and Astronomy, Uppsala University, Box 516, 75120 Uppsala, Sweden. [4] Institute for Applied Materials, Karlsruhe Institute of Technology, Hermann-von-Helmholtz-Platz 1, D-76344 Eggenstein-Leopoldshafen, Germany. [5] Max Planck Institute for Intelligent Systems, Heisenbergstraße 3, D-70569 Stuttgart, Germany. Correspondence and requests for materials should be addressed to J.Z. (email: jzhu@mail.tsinghua.edu.cn).

Since the first demonstration by Schattschneider *et al.*[1] in 2006 that electron magnetic circular dichroism (EMCD) signal can be detected using high-energy electrons in the transmission electron microscope (TEM), theoretical and experimental progress have led to improvements in spatial resolution and signal-to-noise ratio[2–5], an improved fundamental understanding of the technique[6–10] and quantitative measurements of spin and orbital magnetic moments with both element and site specificity[11–14]. In Lorentz mode, with the conventional TEM objective lens (which would normally impart a strong out-of-plane magnetic field to the sample in the electron beam direction) switched off, EMCD in principle allows the intrinsic (remanent) magnetic states of materials to be studied in magnetic-field-free conditions[15]. However, until now the technique has only been able to detect magnetic signals in the out-of-plane (electron beam) direction, in contrast to magnetic characterization methods based on phase contrast in the TEM such as Lorentz microscopy, differential phase-contrast imaging and off-axis electron holography that are sensitive to in-plane components of the magnetic flux density within and around the sample[16,17]. Although electron holographic tomography can in principle be used to visualize three-dimensional magnetic vector fields in materials, it is still a very challenging technique with stringent requirements on sample preparation, microscope operation and data processing[17,18].

In contrast to EMCD, its counterpart, X-ray magnetic circular dichroism, allows magnetic information about sample surfaces to be measured. The recorded X-ray magnetic circular dichroism signal depends on the relative orientations of the helicity of the incident X-rays and the local magnetic moment in the specimen projected onto the propagation direction of the X-rays[19]. As the sample is illuminated by a relatively large beam (typically at least a few tens of nm in size), the magnetization in the sample must be uniform and well defined to avoid signals from neighbouring magnetic domains being averaged out. An external magnetic field is therefore often applied during such experiments and, as for EMCD, the signal is then only recorded from components of the magnetization that are parallel to the beam direction. Actually, the magnetization direction that an EMCD signal is sensitive to is determined by momentum transfer[10,20].

Here, we show that by selecting a specific direction of momentum transfer in the diffraction plane, it is possible to detect in-plane magnetization using EMCD, thereby extending the technique to provide element-specific three-dimensional magnetization characterization in the TEM from samples that are either in their remanent magnetic state or examined in the presence of a chosen applied magnetic field. We use theoretical calculations to propose an operational mode of EMCD that is sensitive to in-plane magnetization and define a diffraction geometry that can be used to separate in-plane from out-of-plane EMCD signals. We use the approach to measure in-plane magnetic signals from a single crystalline Co sample in Lorentz mode, thereby demonstrating a new powerful working principle for the technique.

## Results

**Theoretical background.** An EMCD experiment involves recording two electron energy-loss (EEL) spectra from specific conjugate positions in the diffraction plane. The EMCD signal is then obtained by calculating the difference between these two spectra. In the conventional EMCD technique for detecting out-of-plane magnetization, the EMCD signal can be written in the form[21],

$$\Delta\sigma = K\left(\mu_{+} - \mu_{-}\right)\left(\mathbf{q}\times\mathbf{q}^{'}\right)\mathbf{e}_z \qquad (1)$$

where the dynamical diffraction coefficient $K$ is determined from the diffraction conditions, the intrinsic magnetic circular dichroism signal $\mu_{+} - \mu_{-}$ is determined from the magnetic properties of the material, $\mathbf{e}_m$ is the direction of magnetization (where $m$ represents the direction of magnetization and $m = x$, $y$ or $z$). The vectors $\mathbf{q}$ and $\mathbf{q}'$ denote momentum transfers between the plane-wave components of the Bloch waves. Approximately, $\mathbf{q} = \mathbf{k}_{out} - \mathbf{k}_{in} + \mathbf{h} - \mathbf{g}$ and similarly for $\mathbf{q}'$. $\mathbf{k}_{out}$ and $\mathbf{k}_{in}$ are the incident and outgoing wave vectors, while $\mathbf{g},\mathbf{h}$ are the indices of the plane-wave components (see refs 10,13 for more details).

The direction of magnetization that is detected using EMCD is related to the momentum transfer. In conventional TEM mode, the sample is usually fully saturated magnetically in the $z$ direction (parallel to the electron beam) by the strong magnetic field of the objective lens, as shown in Fig. 1a. Only the $\mathbf{q}_x$ and $\mathbf{q}_y$ components of momentum transfer are then relevant and the final EMCD signal is sensitive to the magnetization in the $z$ direction. In direct analogy, the $\mathbf{q}_x$, $\mathbf{q}_z$ and $\mathbf{q}_y$, $\mathbf{q}_z$ components are sensitive to the components of magnetization in the sample that are oriented in the $y$ and $x$ directions, respectively. In general, the magnetization can be oriented in any direction in Lorentz mode, as shown in Fig. 1b. A diffraction pattern can then contain information about any (or all) of the three components of magnetization[10]. If the distribution of a desired component of the EMCD signal in the diffraction plane is known, then the corresponding signal can be detected by selecting an appropriate diffraction geometry.

Figure 2 shows simulations of the three components of the EMCD signal in the diffraction plane for hexagonally close packed (*hcp*) Co performed for a sample thickness of 20 nm and

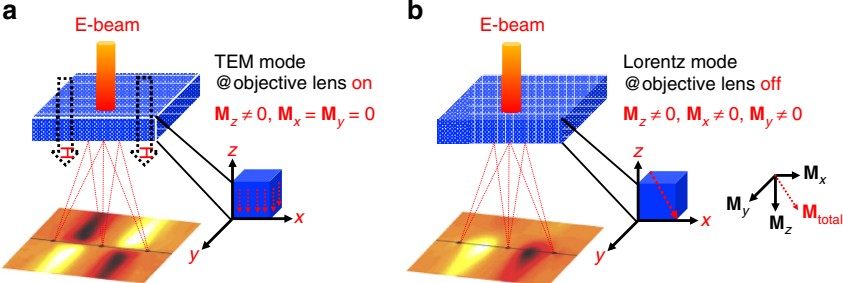

**Figure 1 | Schematic diagrams illustrating out-of-plane and in-plane magnetic measurement using the EMCD technique.** (**a**) Diffraction geometry for the EMCD technique in TEM mode with the objective lens on. The magnetization is saturated in the electron beam (E-beam) direction (the $z$ direction) by the strong applied magnetic field, that is, $\mathbf{M}_x = \mathbf{M}_y = 0$. The EMCD signal is distributed over the four quadrants in the diffraction plane. (**b**) Diffraction geometry for the EMCD technique in Lorentz mode with the objective lens off. In general, the magnetization in each direction ($\mathbf{M}_x$, $\mathbf{M}_y$, $\mathbf{M}_z$) can be non-zero. For illustrative purposes, the diffraction plane shows the distribution of the EMCD signal corresponding to magnetization in the $y$ direction.

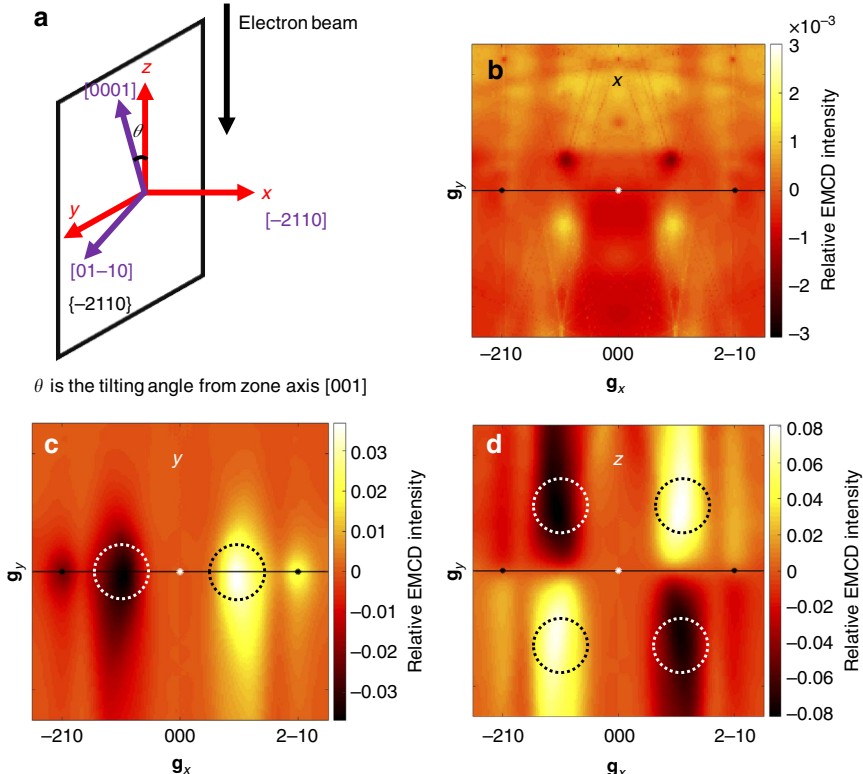

**Figure 2 | Simulation of EMCD signals along three directions.** (**a**) Schematic diagram showing the coordinate system, diffraction geometry and crystallographic orientation for Co in a ( − 210) three-beam orientation. The crystallographic directions ([ − 210], [010] and [001]) are indicated in purple. Miller indices are shown using a four-axis coordinate system for *hcp* Co. By rotating the crystallographic coordinate system in the ( − 210) plane, a ( − 210) three-beam diffraction geometry is achieved. The tilting angle is ∼6°. The *x*, *y* and *z* axes indicate the coordinate system for the diffraction geometry, where *x* is along [ − 210] and *z* is the electron beam direction. These axes correspond to the magnetization directions detected using in-plane and out-of-plane EMCD, respectively. (**b–d**) Simulations of the *x*, *y* and *z* components of the reciprocal space distributions of the EMCD signals in a ( − 210) three-beam geometry for a specimen thickness of 20 nm and an accelerating voltage of 300 kV. The black spots show the positions of the transmitted and diffracted spots. The corresponding diffraction indices are marked along the coordinate axis. White and black circles mark the detector positions.

an accelerating voltage of 300 kV, with the ( − 210) planes strongly excited in a three-beam orientation. Here, we define the systematic reflection direction as the *x* axis, the perpendicular direction as the *y* axis and the electron beam direction as the *z* axis, as shown in Fig. 2a. The simulations shown in Fig. 2b–d confirm that signals originating from the $M_x$, $M_y$ and $M_z$ components are all present in the diffraction plane but have different distributions. The EMCD technique is therefore in principle sensitive to both in-plane and out-of-plane magnetization. It should be noted at this stage that experimental EMCD signals are formed from two terms that are multiplied by each other: the relative distribution of intensity of the EMCD signal and the intrinsic magnetization ($\mu_+ - \mu_-$). If the magnetization in any direction is zero then the corresponding EMCD signal will also be zero, just as in the conventional TEM mode of EMCD, for which in general $M_x = 0$ and $M_y = 0$.

The signal in the *z* direction is distributed between the four quadrants just as for conventional EMCD, in an antisymmetric arrangement relative to the *x* and *y* axes. The detector can then be placed in the first (second) and fourth (third) quadrants. In contrast, the EMCD signal in the *y* direction stays primarily on the *x* axis, with only antisymmetry relative to the *y* axis. The detector positions should therefore be symmetrical relative to the *y* axis. The EMCD signal in the *x* direction is very different from that in the *y* and *z* directions. It is symmetrical relative to the *y* axis, but there is no symmetry relative to the *x* axis. A signal recorded from detector positions that are symmetrical relative to

the left and right half diffraction planes therefore cancels out and it is not possible to obtain an EMCD signal in the *x* direction.

The EMCD signal also differs in intensity between the three components, as it depends on the relative magnitudes of the *x*, *y* and *z* components of momentum transfer. The $q_y$ component originates primarily from the placement of the detector relative to the transmitted beam. For $q_x$, not only the contribution from the detector placement, but also the strength of the excited Bragg spots contributes additional momentum transfer to $q_x$ in the systematic row direction. For $q_z$, the primary origin is the energy loss of several hundreds of eV, whose contribution is rather small. The products $q \times q'$ that contain $q_x$ are therefore expected to produce the strongest signals, as shown in Fig. 2c,d for the *y* and *z* directions. In contrast, the *x* component of the EMCD signal will be weak, as shown in Fig. 2b, as only $q_y$ and $q_z$ are relevant.

The simulations for Co shown above were performed separately for the three components (*x*, *y* and *z*). However, the true signal will be a superposition of the three distributions that overlap with each other in the diffraction plane. Their separation is then crucial. In some special cases, the situation is very simple, such as when $M_x = M_y = 0$ in conventional TEM mode in the presence of the ∼2 T magnetic field of the objective lens that is likely to fully saturate the sample in the *z* direction. In contrast, strong shape anisotropy and demagnetization energy often stabilize the magnetization in the plane of the TEM specimen in Lorentz mode, resulting in a negligible EMCD signal in the *z* direction. To measure the *y* component of the EMCD signal, the

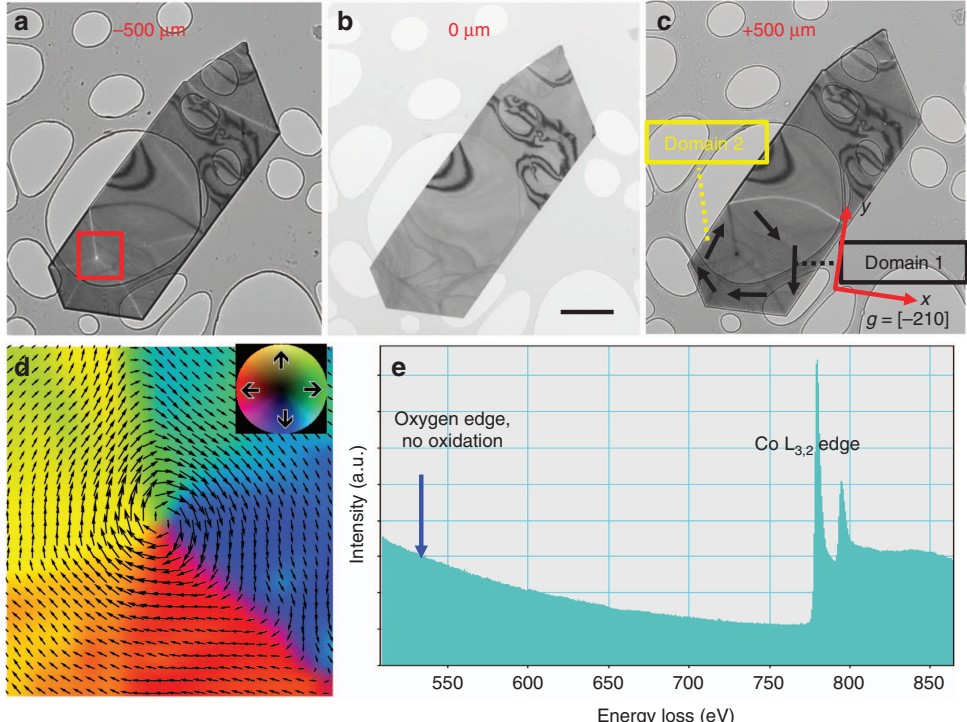

**Figure 3 | Lorentz TEM results acquired from a Co nanoplate on a holey C support film.** (**a**–**c**) Fresnel defocus images recorded (**a**) underfocus, (**b**) in focus and (**c**) overfocus. The defocus values $\Delta z$ used were $\pm 500\,\mu m$. The area marked by a red box in **a** was used for analysis using the transport of intensity equation (TIE). The crystallographic $x$ ([$-210$]) and $y$ axes of the Co nanoplate are marked in **c**. The black arrows in **c** show the direction of the local projected magnetic induction determined using TIE analysis, as shown in **d**. The black scale bar denotes $5\,\mu m$. The black and yellow boxes in **c** mark two magnetic domains with opposite magnetic induction vectors. (**d**) Magnetic vector map determined from the marked region in **a** using TIE analysis. (**e**) EEL spectrum recorded from the Co nanoplate, showing no oxide peak and suggesting metallic Co. The blue arrow marks the expected position of the O-K edge ($\sim 530\,eV$).

detector can be placed in the left and right half diffraction planes where the intensity is strong, as shown in Fig. 2c. However, a suitable diffraction geometry cannot be chosen to measure the $x$ component simultaneously. Instead, an alternative three-beam geometry can be selected to measure the $x$ component by exciting a perpendicular systematic row (see the Discussion section below). For some materials that have strong shape or magneto-crystalline anisotropy, there may be a component of magnetization in the $z$ direction in Lorentz mode. The antisymmetric character of the $z$ component in the up-down half diffraction plane[22,23] can then be utilized (see Supplementary Note 1 and Supplementary Fig. 1), with the detector placed on the $x$ axis to cancel out the EMCD signal from the $z$ direction, as shown by a circle in Fig. 2c. By placing the detector on the $x$ axis, it is possible to extract the in-plane EMCD signal (in the $y$ direction) separately, even in the presence of an additional $z$ component.

**Experimental results**. We studied a single crystalline Co nano-plate that had a width of $10\,\mu m$, a length of $30\,\mu m$ and a uniform thickness of $20\,nm$ to verify the theoretical predictions described above. Details about how the specimen was prepared are given in the Methods section. The geometry of the specimen ensures an in-plane remanent magnetization distribution because of shape anisotropy. Figure 3a–c shows Lorentz TEM images recorded from the Co nanoplate in magnetic-field-free conditions. The transport of intensity equation (TIE)[24] was applied to the images that show magnetic domain walls and a magnetic vortex structure to obtain an independent measurement of the magnetization distribution in the specimen. Figure 3d shows a map of the

projected in-plane magnetization in the specimen inferred from the TIE-derived phase.

The [$-210$] crystallographic direction of the Co nanoplate corresponds to the $x$ axis, that is, to the systematic reflection direction used in the EMCD experiments. The $y$ axis is perpendicular to this direction, as shown in Fig. 3c. The lack of an O-$K$ edge in the EEL spectrum shown in Fig. 3e confirms the absence of detectable oxidation of the Co nanoplate. For in-plane EMCD measurements, we chose two domains, which are labelled domain 1 and domain 2, magnetized almost parallel and antiparallel to the $y$ direction and marked using black and yellow boxes in Fig. 3c. A change in the sign of the in-plane EMCD signal is expected between the domains, based on the change in in-plane magnetization direction.

We first performed EMCD experiments in TEM mode in the presence of the strong ($\sim 2\,T$) magnetic field of the microscope objective lens, with the sample saturated magnetically in the $z$ direction and the ($-210$) planes strongly excited in a three-beam geometry. The two magnetic domains that were previously observed in Lorentz mode were now examined with the detector placed in symmetrical positions in the first and fourth quadrants, as shown in Fig. 4a,b with the noise level of $\sim 0.27\%$, $\sim 0.28\%$ (it is relative to the peak intensity of spectra at $L_3$ edge) by evaluating the noise fluctuations in the pre-edge edge of EMCD spectra, respectively. (The EMCD signals in Fig. 4a,b are obtained by using the double difference method[11] that involves acquiring EEL spectra from all four quadrants.) The EMCD signal in the $z$ direction was measured easily, with the two signals exhibiting almost the same relative intensities because of the same diffraction condition and magnetic properties of the specimen

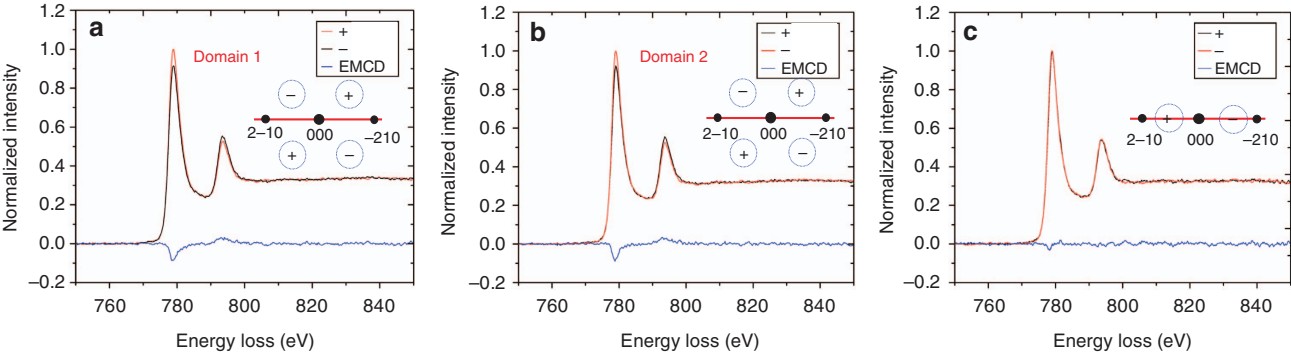

**Figure 4 | Experimental EMCD signals recorded from a Co nanoplate in different diffraction geometries in the fully saturated TEM mode.** (**a,b**) EMCD signals recorded in a three-beam condition at the positions of domains 1 and 2, respectively. (**c**) EMCD signal recorded in an in-plane diffraction geometry in TEM mode. The black and red lines correspond to EEL spectra recorded from the '+' and '−' positions, respectively, while the blue lines are the resulting EMCD signals. The schematic diagrams in each figure represent the diffraction geometries. The blue circles show the positions of the EELS entrance aperture.

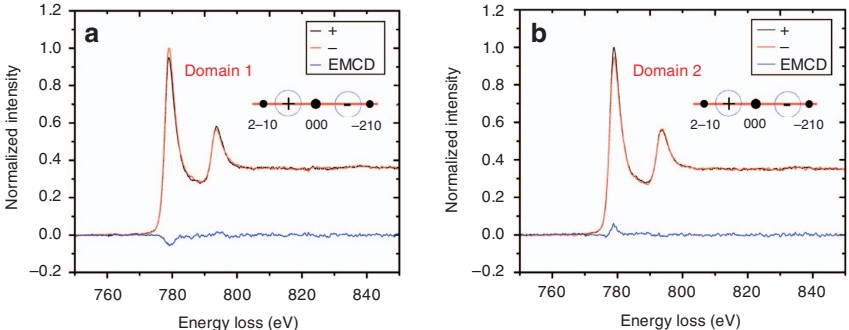

**Figure 5 | Experimental in-plane EMCD signals recorded from a Co nanoplate in the Lorentz mode.** (**a,b**) Experimental EMCD signals acquired for the three-beam case from domains 1 and 2, respectively, in the in-plane diffraction geometry in Lorentz mode. The diffraction conditions were the same as for the TEM experiment shown in Fig. 4.

used for the two measurements. In contrast, when the detector was placed on the $x$ axis in the in-plane EMCD geometry (see Fig. 2c), no measurable EMCD signal was detected, as shown in Fig. 4c with the noise level of ∼0.30%. This result is expected because the in-plane magnetization component is zero when the microscope objective lens is strongly excited.

We performed in-plane EMCD measurements from the same region in Lorentz mode in magnetic-field-free conditions after returning the Co nanoplate to the remanent magnetic state shown in Fig. 3 (see Supplementary Fig. 6). The in-plane EMCD geometry was applied to the two domains and EEL spectra were acquired along the $x$ axis. Figure 5 shows the measured EMCD signals with the noise level of ∼0.34% and ∼0.34%, respectively. The signals reverse in sign between the two domains. These results directly confirm the theoretical proposal that in-plane magnetization can be detected in the new EMCD geometry. We note that the in-plane signal recorded in Lorentz mode is smaller than the out-of-plane signal recorded in TEM mode, even though the magnetic moment of Co in the detection direction is the same, because of the smaller momentum transfer $\mathbf{q}_z$, as shown in Fig. 2b,c. Here, we provide a quantitative estimate of the strength of the EMCD signal from the experimental spectra. For the $z$ component in the out-of-plane geometry shown in Fig. 4a,b, the strength of EMCD signal for the $L_3$ edge is estimated to be 8.3 ± 0.4% and 8.4 ± 0.4%, respectively. For the in-plane $y$ component in the Fig. 5a,b, the strength is 4.8 ± 0.9% and 5.0 ± 0.8%, respectively. According to the theoretical simulations shown in Fig. 2, the relative intensity in the diffraction plane is

obtained by averaging the value in the entrance aperture. The relative intensity is ∼5.0 and 11.0% for the in-plane $y$ component and the out-of-plane $z$ component, respectively, that are close to our experimental results.

## Discussion

In our experiments, we only measured an in-plane EMCD signal in the $y$ direction, as the signal in the $x$ direction is symmetrical relative to the $y$ axis and it is not possible to design a geometry that can be used to extract the $x$ and $y$ signals in a single three-beam diffraction geometry. However, as discussed above, it is possible to use theoretical simulations to design a suitable diffraction geometry for detecting the EMCD signal along any crystallographic direction. For example, in the present study, a signal along the $x$ axis, that is, [−210], can be measured by changing the diffraction geometry to strongly excite the (010) planes (see Supplementary Fig. 2) by tilting the specimen by a small angle (a few degrees). The in-plane EMCD technique is therefore applicable to two-dimensional in-plane magnetic measurements.

The nanoplate of *hcp* Co that we studied was orientated along the [001] direction that is the crystallographic easy axis of magnetization. In magnetic-field-free conditions, the shape anisotropy of the nanoplate forces the magnetization vector to lie in-plane to decrease the demagnetization energy. The out-of-plane geometry was also used to detect the magnetization in the $z$ direction in domains 1 and 2 in Lorentz mode

(see Supplementary Fig. 3). Although the signal for the in-plane $y$ direction is not symmetrical with respect to the conjugate positions in the first and fourth quadrants, this difference is not evident after normalization. The absence of a measurable signal in Supplementary Fig. 3 confirmed that there was no detectable out-of-plane magnetic component in Lorentz mode. The objective lens was then excited in free lens mode to apply a magnetic field in the electron beam direction. In the presence of applied fields of 400 and 833 mT, the magnetic domain walls disappeared and the specimen adopted a saturated single domain state. The EMCD signal in the $z$ direction was then detected with almost the same relative intensity as in conventional TEM mode (see Supplementary Fig. 4), confirming that the sample was fully saturated in the presence of the applied magnetic field. In this way, by using both conventional and in-plane EMCD, the signal in different directions could be measured in Lorentz mode.

Although the EMCD signals that are described here were acquired in a three-beam geometry, all other experimental setups that have been proposed for conventional EMCD, including energy-filtered TEM[11], scanning TEM[3,25] and convergent beam electron diffraction[2], can also be applied to in-plane EMCD to achieve high spatial resolution, high signal-to-noise ratio and quantitative measurements. A recently developed zone axis diffraction geometry also promises to provide new opportunities for in-plane EMCD measurements[26] by allowing the three magnetic components to be extracted simultaneously from a single diffraction pattern[10]. The present experiments are conducted in Lorentz-TEM mode at medium resolution. The Lorentz-scanning TEM mode could be used in the future to achieve much higher spatial resolution. An assessment of the sampled amount of material versus signal-to-noise ratio, as reported in ref. 27, could then be used to provide an estimate of the resolution limit. Theoretical analysis should then also be performed to discuss the impact of overlapped diffracted discs on the in-plane EMCD signals which are just localized on the systematic reflection axis. In the present study, these sum rules[8,9] could not be applied to the measurement of quantitative magnetic parameters because distortions were present in the diffraction pattern even in $C_S$-corrected Lorentz mode, resulting in the need for an improved diffraction alignment for such experiments in the future.

In summary, we have successfully demonstrated an in-plane EMCD technique that is based on a specific diffraction geometry and can be used to measure the magnetization distribution in a specimen perpendicular to the electron beam direction. We obtained experimental measurements of in-plane EMCD signals from magnetic domains in a Co nanoplate in magnetic-field-free conditions in Lorentz mode. Our results are consistent with theoretical calculations and demonstrate that both in-plane and out-of-plane magnetic properties of nanoscale materials can be studied using EMCD, paving the way for new areas of research on magnetism at the nanometre scale.

## Methods

**Sample fabrication.** The Co nanoplates used in this work were obtained using molecular beam epitaxy deposition onto Mo and W wires with diameters of $\sim 0.3$ mm. Synthesis of the nanoplates was carried out in an ultra-high vacuum molecular beam epitaxy chamber with a base pressure of below $2 \times 10^{-10}$ mbar. The substrate temperature for Co whisker growth was $\sim 760\,°C$. The typical deposition rate was $\sim 0.05$ nm s$^{-1}$, monitored using a quartz balance. The nominal thickness of the deposited Co is 180 nm. The specimen plate was constantly rotated around its surface normal during growth to achieve a more homogeneous deposition. The nanoplates were then scratched from the substrate surface using a TEM grid for TEM investigations.

**Theoretical calculations.** Simulations were performed using the modified automatic term selection (MATS) software, with accurate summation over Bloch waves and their plane wave components[28]. The algorithm is optimized to avoid

summation of negligible terms and to improve scaling of the summation. The simulations are based on a supercell with an orthogonal coordinate system (see Supplementary Note 2 and Supplementary Fig. 7). The cutoff criterion was set to $P_{min} = 0.0001$ during the calculations to ensure convergence. The number of beams generated for the eigenvalue problem was 97/104 of the incident and outgoing Bloch waves for the $(-210)$ three-beam case. Approximately 6° of specimen tilt from the [001] zone axis was used to reach a $(-210)$ three-beam condition in the simulation, as in the experimental conditions. The specimen thickness was 20 nm and the accelerating voltage was 300 kV in the simulations.

**Experimental data acquisition and processing.** All of the experiments were carried out using the FEI Titan PICO TEM operated at 300 kV. This microscope, which is equipped with a post-column energy filter (Gatan Quantum 966 ERS)[29], provided stable conditions for the experiments. Lorentz TEM and in-plane EMCD experiments were performed in Lorentz mode, with the conventional microscope objective lens switched off and the specimen in magnetic-field-free conditions ($<0.1$ mT). Chosen magnetic fields in the electron beam direction were applied by partially exciting the objective lens in Lorentz mode using free lens control. Out-of-plane EMCD experiments were performed in conventional TEM mode with the objective lens fully excited and a magnetic field of $\sim 2$ T applied to the sample.

For the EMCD measurements, a nearly parallel beam with a probe diameter of $\sim 50$ nm was used to illuminate the sample. The specimen tilt angle was chosen to be $\sim 6°$ to reach the $(-210)$ three-beam diffraction orientation from the [001] zone axis. There is a slight difference in orientation between these two magnetic domains because of bending of the sample. We made separate slight orientation adjustments for the two regions during the experiments to achieve precise three-beam case. The acquisition time for each spectrum was a few seconds to achieve a high signal-to-noise ratio with negligible irradiation damage. Pre-edge background subtraction was performed for all EEL spectra. Then, an energy calibration is first applied to the 'plus' and 'minus' spectra to correct for energy drift during the experiments. The energy dispersion is 0.2 eV per channel in the experiments. All of the spectra are subdivided to have an energy dispersion of 0.05 eV per channel to improve the accuracy of the energy calibration. After that, the '+' and '−' EELS edges were normalized by the integrated intensity present in the region of the post-edge energy window between 50 and 100 eV after the onset energy of the $L_3$ edge to minimize any effects of asymmetry, based on the assumption that in this energy range the magnetic signal is negligible and only nonmagnetic spectral components remain[30]. The desired EMCD signal was obtained by subtracting the two final spectra.

TIE analysis was based on the acquisition of three Lorentz TEM images: underfocus, in focus and overfocus. Phase images, which were used to infer the local projected in-plane magnetic induction, were determined from the Lorentz images by solving the TIE using a Matlab code. On the assumption of a constant specimen thickness, a constant in-plane magnetization in the electron beam direction and negligible stray magnetic fields, the gradient of the measured phase shift is directly proportional to the in-plane component of the magnetization in the specimen.

**Data availability.** The data that support the findings of this study are available from the corresponding author on request.

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

## Acknowledgements

This work was financially supported by the Chinese National Natural Science Foundation (11374174, 51390471, 51527803), the National Key Research and Development Program (2016YFB0700402) and the National 973 Project of China (2015CB654902). The work made use of the resources of the Ernst Ruska-Centre for Microscopy and Spectroscopy with Electrons in Jülich and the National Center for Electron Microscopy in Beijing. The research leading to these results received funding from the European Research Council under the European Union's Seventh Framework Programme (FP7/2007-2013)/ERC grant agreement number 320832. J.R. acknowledges Swedish Research Council and Göran Gustafsson's Foundation for financial support. We thank Dr Xiaoyan Zhong from Tsinghua University for helpful discussions.

## Author contributions

J.Z. and D.S. proposed the work and organized the team for the project. D.S. and J.R. performed the theoretical calculations. D.S., A.H.T., Z.-A.L. A.K. and R.E.D.-B. conducted the experiments. D.S. analysed the data. W.H. and G.R. prepared the Co nanoplates. Z.-A.L. proposed the sample for the experiments. D.S. and J.Z. wrote the manuscript. All of the authors discussed the results and commented on the manuscript.

## Additional information

**Competing interests:** The authors declare no competing financial interests.

