## [Peer Review File · Nature Communications]

Reviewer #1 (Remarks to the Author):

This is an interesting original paper. The central message, namely that in-plane fields can be measured with this geometry appears to be correct, but I have some reservation on two details:

1) in Fig 2b the EMCD signal corresponding to an x-component of magnetisation should vanish along the systematic line (on the very x axis) because there, the vector product $q \times q'$ has only a y component, the inner product of which with $e_x = 0$. Also, its strength appears too high (ca 0.02 as compared to 0.03 for the y component), contrary to what the authors state in the text. Authors should carefully check the simulation and in case it is correct explain why there is intensity on the x axis.

2) in Fig. 5, the experimental EMCD signal (in this case the y component) is about 10% , equally in strength as the z component in out-of-plane geometry (Fig. 4a). This is very questionable already from basic reasoning that suggests a much smaller signal (as authors confirm in the text), and also from what follows from simulations in Fig.2 where it seems to be about 3 %. Authors should carefully check if there isnt an artifact responsible for this high signal.

3) with reference to the previous problem, authors should provide an error estimate in the experimental spectra, ideally error bars.

If these two problems can be solved the paper can be published. In this case, there are some additional points that should be clarified:

- line 50: a surface in the strict sense has no depth
- line 79f: this is not the DDSCS but already its difference
- line 83: better use e_m or similar, not e_z when the magnetisation is not necessarily in z direction.
- line 84: the momentum transfers q and q' are neither defined in the text nor in the figures
- authors use "left, right, up, down diffraction plane". More precisely, these are half planes, and what is meant should be explained..
- line 125 ff: this part is confusing. q_x and q_y both depend on the detector placement.
- line 166f: wording: "[...] was used to establish the [-210] direction as x-axis".
- line 181: why did authors not use all 4 quadrants as suggested for symmetric n-beam cases? This would reduce artefacts from the 6 degree tilt as shown by J. Ruzs.
- line 240: "These sum rules..." Sum rules were never mentioned before.
- Fig. 2a: the triple plane is confusing. We are in diffraction, so one plane would be clearer.
- Fig 3: from bent contours it seems that the specimen is considerably bent with different orientations in domains 1 and 2. Was it separately reoriented before measurements. If not, what are the uncertainties in orientation ?
- supplement:
- line 21: the A comes as a surprise. It would be easier to grasp when the notation from the main text were used.
- Fig S1 caption: is it z or y and z ?
- Fig S2: for which magnetisation? Was it assumed to be saturated in each of x,y, and z separately?

Reviewer #2 (Remarks to the Author):

The authors of this manuscript demonstrate a novel method to extend EMCD measurements, the counterpart of XMCD, to magnetisation in 3 dimension by exploiting the detection geometry in the diffraction plane. They show both theoretically and experimentally that the method is viable and show that this considerably widens the scope of chemically selective magnetic measurements in the TEM at medium resolution. The paper is exceptionally well written and only few minor comments need to be addressed:

-one could argue that the term EMCD with C standing for circular or chiral is still justified in this setup. Indeed the in-plane components will lead to a phase signature of the inelastic electrons that will resemble a pi-orbital rather than the typical vortex state that is involved with an out of plane magnetic excitation. The term EMLD with L for linear might be more appropriate.

-an estimate of the sensitivity of this method with respect to strain would be important to verify if small amounts of strain could not lead to spurious measurement. Moreover this strain could couple to the magnetic state, making the interpretation even more convoluted.

-an evaluation of the total sampled amount of material versus signal to noise ratio could provide an estimate of the application limits of this technique and of the fundamental limits avoiding this method to go to atomic resolution mapping.

-the paragraph from line 123 to 131 is correct but could be explained in a clearer way.

-shifting the diffraction geometry during the experiment is in practice cumbersome. Are there other ways to selectively measure x and y component of the magnetization separately?

-is there a reason why the authors didn't perform an energy filtered diffraction experiment revealing the full details of the scattering to be compared to the theory. If I understand correctly the experiment only verifies a few fixed positions in this pattern which still could mask severe differences between experiment and theory.

-line 305 'home-written' seems to imply that due to lack of time at the lab, the software needed to be written from home.

Reviewer #1:

This is an interesting original paper. The central message, namely that in-plane fields can be measured with this geometry appears to be correct, but I have some reservation on two details:

(1) in Fig 2b the EMCD signal corresponding to an x-component of magnetization should vanish along the systematic line (on the very x axis) because there, the vector product $\mathbf{q} \times \mathbf{q}'$ has only a y component, the inner product of which with $\mathbf{e}_x = 0$. Also, its strength appears too high (ca 0.02 as compared to 0.03 for the y component), contrary to what the authors state in the text. Authors should carefully check the simulation and in case it is correct explain why there is intensity on the x axis.

Answer:

We thank the referee for pointing out this inconsistency. We have consulted this issue with Jan Ruzs, who has now become a co-author of this manuscript. In the submitted version of the paper, we did not realize that the coordinate system for diffraction pattern and coordinate system for individual spin components are not necessarily the same and that in our geometry it was crucial to transform the maps into the same coordinate system as for the diffraction pattern. After the necessary transformation matrix had been applied, the magnitude of the magnetic signal for magnetization along x-direction decreased by an order of magnitude (Fig. R1). We double-checked this result using an orthogonal supercell (Fig. R2), for which such transformation is not necessary, and we obtained numerically the same result (Fig. R3) as in the revised figure (Fig. R1). We apologize for this error and thank once more the referee for pointing this out. We have added these information in the **Method** part, “**theoretical calculations**”, as well as in the Supplementary Information, in our revised manuscript.

Figure R1. Corrected simulated results for *hcp* Co. From left to right, they show respectively the *x*, *y* and *z* components of the reciprocal space distributions of the EMCD signals in a (-210) three beam geometry for a specimen thickness of 20 nm.

Figure R2. (a) Primary cell with 4*4 unit cells projected along the (001) direction. (b) Supercell used for simulations, taken from the red rectangle box in (a).

Figure R3. Simulated results for *hcp* Co, obtained by taking the supercell with an orthogonal coordinate system as shown above in Figure R2. From left to right, the images show the *x*, *y* and *z* components, respectively, of the reciprocal space distributions of the EMCD signals in a (-210) three beam geometry for a specimen thickness of 20 nm.

Revision:

(1) We have replaced Fig. 2 with our new simulated results, which are based on a supercell with an orthogonal coordinate system.

(2) In the Method part, “theoretical calculations”, we add the following part.

The simulations are based on a supercell with an orthogonal coordinate system (see supplementary information, Fig. S7).

(3) In the Supplementary Information, we have added the following part to describe the supercell that we used in our simulations,

The supercell for *hcp* Co with an orthogonal system is shown in Fig. S7. The simulations are based on this supercell. The (100) planes in the supercell correspond to (-210) planes in the primary unit cell, while the (010) planes correspond to (010) planes in the primary unit cell.

Figure S7. (a) Primary cell with 4*4 unit cells projected along the (001) direction. (b) Supercell used for simulations, taken from the red rectangle box in (a).

(2) in Fig. 5, the experimental EMCD signal (in this case the y component) is about 10%, equally in strength as the z component in out-of-plane geometry (Fig. 4a). This is very questionable already from basic reasoning that suggests a much smaller signal (as authors confirm in the text), and also from what follows from simulations in Fig.2 where it seems to be about 3%. Authors should carefully check if there isn't an artifact responsible for this high signal.

(3) with reference to the previous problem, authors should provide an error estimate in the experimental spectra, ideally error bars.

Answer:

We have checked all the experimental data and processing steps. The EEL spectra acquired in the experiments have an energy dispersion of 0.2 eV/channel. As energy drift is always present during signal acquisition between the “plus” and “minus” positions, an energy calibration is needed before the subtraction for EMCD signal. In

our previous works, we always subdivided the energy dispersion and use 0.1 eV/channel during the alignment of different spectra and it works very well. Therefore, we also used 0.1 eV/channel in our original manuscript. However, the referee’s comment has drawn our attention to an artifact in Fig. 5(a). There is still a misalignment of ~ 0.05 eV in Fig. 5(a) in our original manuscript, as shown below in Fig. R4. Therefore, we subdivided the energy dispersion with 0.05 eV/channel to perform the alignment again. The misalignment in the corrected spectra was then almost negligible, as shown below in Fig. R4. In consequence, the peak strength of EMCD signal has decreased. We also rechecked all other spectra using the energy dispersion of 0.05 eV/channel and found that no obvious changes were observed. The results in our revised manuscript were updated and this detail of the data processing was added to the **Methods** part, “**Experimental data acquisition and processing**” in our revised manuscript.

Figure R4. The upper figure is the same as Fig. 5(a) in our original manuscript with ~ 0.05 eV misalignment. The lower figure is an updated version of Fig. 5(a) after a new energy calibration performed using an energy dispersion of 0.05 eV/channel.

With regard to the strength of experimental spectra, we provide a quantitative estimate of the strength of the L_3 edge in the EMCD signal from the experimental spectra. For the z component in the out-of-plane geometry in Figs 4(a) and 4(b), the strength of the EMCD signal is estimated to be $8.3\% \pm 0.4\%$ and $8.4\% \pm 0.4\%$

respectively. For the in-plane y component in the updated Figs 5(a) and 5(b), the strength of the signal is estimated to be $4.8\% \pm 0.9\%$ and $5.0\% \pm 0.8\%$ with the error bars as suggested by referee, respectively.

According to theoretical simulations in the updated version of Fig. 2, the relative intensity of the EMCD signal is obtained by averaging the value in the entrance aperture. The relative intensity is approximately 5.0% and 11.0% for the in-plane y component and the out-of-plane z component, respectively, which are close to experimental results. The relative intensity is doubled here as the signal obtained from the subtraction of “plus” and “minus” positions, with the value of 2.5% and 5.5% for y and z component in one detector position, respectively.

We also give an estimate of noise by evaluating the noise fluctuations in the pre-edge edge of EMCD spectra. The noise is for Figs 4(a), 4(b) and 4(c) are $\sim 0.27\%$, $\sim 0.28\%$ and $\sim 0.30\%$ (it is relative to the peak intensity of spectra at L_3 edge), respectively. The noise for Figs 5(a) and 5(b) are $\sim 0.34\%$ and $\sim 0.34\%$, respectively. This information has added to the revised manuscript.

Revision:

(1) Figure 5(a) has been updated after the new energy calibration.

(2) We have described the energy calibration in the data processing section of the Methods part, “Experimental data acquisition and processing”, in our revised manuscript, as follows,

An energy calibration is firstly applied to the “plus” and “minus” spectra to correct for energy drift during the experiments. The energy dispersion is 0.2 eV/channel in the experiments. All of the spectra are subdivided to have an energy dispersion of 0.05 eV/channel to improve the accuracy of the energy calibration.

(3) A discussion of the EMCD signal strength in the experiments and simulations has been added to the last paragraph of “Experimental results” part of our revised manuscript, as follows,

Here, we provide a quantitative estimate the strength of the EMCD signal from the experimental spectra. For the z component in the out-of-plane geometry shown in Figs

4(a) and 4(b), the strength of EMCD signal for the L_3 edge is estimated to be $8.3\% \pm 0.4\%$ and $8.4\% \pm 0.4\%$, respectively. For the in-plane y component in the Figs 5(a) and 5(b), the strength is $4.8\% \pm 0.9\%$ and $5.0\% \pm 0.8\%$, respectively. According to the theoretical simulations shown in Fig. 2, the relative intensity in the diffraction plane is obtained by averaging the value in the entrance aperture. The relative intensity is approximately 5.0% and 11.0% for the in-plane y component and the out-of-plane z component, respectively, which are close to our experimental results.

(4) The noise level has been added to the revised manuscript.

If these two problems can be solved the paper can be published. In this case, there are some additional points that should be clarified:

- line 50: a surface in the strict sense has no depth

Revision:

In our previous manuscript,

In contrast to EMCD, its counterpart, X-ray magnetic circular dichroism (XMCD), allows magnetic information about sample surfaces to be measured, albeit with limited depth resolution.

In our revised manuscript,

In contrast to EMCD, its counterpart, X-ray magnetic circular dichroism (XMCD) with surface sensitivity, allows magnetic information about sample surfaces to be measured.

- line 79f: this is not the DDSCS but already its difference

Revision:

In our previous manuscript,

The differential scattering cross-section (DDSCS) at opposite ('+' and '-') chiral positions can be written in the form

In our revised manuscript,

The EMCD signal can be written in the form

- line 83: better use e_m or similar, not e_z when the magnetisation is not necessarily in z direction.

Revision:

“ e_z ” has been replaced with “ e_m (where m represents the direction of magnetization and $m = x, y$ or z)”

- line 84: the momentum transfers q and q' are neither defined in the text nor in the figures

Revision:

We have added the following part in our revised manuscript.

The vectors q, q' denote momentum transfers between the plane-wave components of the Bloch waves. Approximately, $q = k_{out} - k_{in} + h - g$ and similarly for q' . k_{out} and k_{in} are the incident and outgoing wave vectors, while g, h are the indices of the plane-wave components (see Refs. 10 and 13 for more details) .

- authors use "left, right, up, down diffraction plane". More precisely, these are half planes, and what is meant should be explained.

Revision:

The description of the diffraction plane has been revised as follows: “left half diffraction plane”, “right half plane”, “upper half plane” and “lower half plane”.

- line 125 ff: this part is confusing. q_x and q_y both depend on the detector placement.

Answer:

Both q_x and q_y indeed depend on the detector placement. As $q = k_{out} - k_{in} + h - g$, there is also a significant contribution to q_x from the excited Bragg spots in the systematical reflection case.

Revision:

We have rewritten this paragraph as follows,

The EMCD signal also differs in intensity between the three components, as it depends on the relative magnitudes of the x , y and z components of momentum transfer. The q_y component originates primarily from the placement of the detector relative to the transmitted beam. For q_x , not only the contribution from the detector placement, but also the strength of the excited Bragg spots contributes additional momentum transfer to q_x in the systematic row direction. For q_z , the primary origin is the energy loss of several hundreds of eV, whose contribution is rather small. The products $\mathbf{q} \times \mathbf{q}'$ that contain q_x are therefore expected to produce the strongest signals, as shown in Figs 2(c) and 2(d) for the y and z directions. In contrast, the x component of the EMCD signal will be weak, as shown in Fig. 2(b), as only q_y and q_z are relevant.

- line 166f: wording: "[...] was used to establish the [-210] direction as x -axis".

Revision:

In our previous manuscript,

A diffraction pattern was used to establish that the [-210] crystallographic direction of the Co nanoplate corresponds to the x axis.

In our revised manuscript,

The [-210] crystallographic direction of the Co nanoplate corresponds to the x axis.

- line181: why did authors not use all 4 quadrants as suggested for symmetric n-beam cases? This would reduce artefacts from the 6 degree tilt as shown by J. Rusz.

Answer:

As the referee suggests, the artefacts can be reduced by using the double difference method, e.g. by acquiring the signal from four quadrants. We do in fact record the signal from all four quadrants and the results presents in Figs 4(a) and 4(b) are final results after applying the double difference method to reduce the artefacts. We only refer to the first and fourth quadrant for two detector positons in our previous manuscript, so as to be able to compare with the signal from the in-plane direction,

which is also only obtained from two detector positions, e.g. the left and right half plane. We apologise that these details were not clearly presented.

Revision:

We have added the following details to the third paragraph of the “Experimental results” part of our revised manuscript,

The EMCD signals in Figs 4(a) and 4(b) are obtained by using the double difference method [11], which involves acquiring EEL spectra from all four quadrants.

[11] Lidbaum, H. et al. Quantitative magnetic information from reciprocal space maps in transmission electron microscopy. Phys. Rev. Lett. 102, 037201 (2009).

- line 240: "These sum rules..." Sum rules were never mentioned before.

Revision:

We have added the reference [8,9] here for the sum rules.

[8] Calmels, L. et al. Experimental application of sum rules for electron energy loss magnetic chiral dichroism. Phys. Rev. B 76, 060409 (2007).

[9] Ruzs, J., Eriksson, O., Novák, P. & Oppeneer, P. M. Sum rules for electron energy loss near edge spectra. Phys. Rev. B 76, 060408 (2007)

- Fig. 2a: the triple plane is confusing. We are in diffraction, so one plane would be clearer.

Revision:

As the referee states, we are in one diffraction plane. We have modified Fig. 2(a) and kept only one diffraction plane.

- Fig 3: from bent contours it seems that the specimen is considerably bent with different orientations in domains 1 and 2. Was it separately reoriented before measurements. If not, what are the uncertainties in orientation?

Answer:

There is a slight difference in orientation between the regions that contain these two magnetic domains because of bending of the sample. We therefore made a slight

adjustment separately for these two regions to achieve the precise three-beam case before measurements during the experiments.

Revision:

We have added the following sentences to our revised manuscript in the Methods part, “Experimental data acquisition and processing”, as follows,

There is a slight difference in orientation between these two magnetic domains because of bending of the sample. We made separate slight orientation adjustments for the two regions during the experiments to achieve precise three-beam case.

- supplement:

- line 21: the A comes as a surprise. It would be easier to grasp when the notation from the main text were used.

Revision:

We have added the notation for A in the revision Supplementary Information.

A is defined as the summation of the coefficient from the upper and lower half diffraction plane for the EMCD signal of z component.

- Fig S1 caption: is it z or y and z ?

Answer:

We have modified the caption of Fig. S1 to make this clear.

Revision:

Figure S1. (a) EMCD signal for the y component under (-210) three-beam case; (b) Summation of the coefficient from the upper and lower half diffraction plane for the EMCD signal of z component (as defined with A in the text).

- Fig S2: for which magnetisation? Was it assumed to be saturated in each of x, y , and z separately?

Answer:

It is assumed that the magnetization is saturated in each direction separately. We have modified the caption of Fig. S2 in our revised Supplementary Information.

Revision:

Figure S2. Simulated reciprocal space distributions of the EMCD signal for the x , y and z components in an (010) three-beam geometry for a specimen thickness of 20 nm and an accelerating voltage of 300 kV. For each component, it is assumed that the magnetization is saturated in this direction in the simulations. The black and white spots mark the positions of the transmitted and diffracted beams in the diffraction plane. The indices of the beams are marked along the coordinate axis. The white and black circles mark the detector positions.

Reviewer #2:

The authors of this manuscript demonstrate a novel method to extend EMCD measurements, the counterpart of XMCD, to magnetization in 3 dimension by exploiting the detection geometry in the diffraction plane. They show both theoretically and experimentally that the method is viable and show that this considerably widens the scope of chemically selective magnetic measurements in the TEM at medium resolution. The paper is exceptionally well written and only few minor comments need to be addressed:

(1) one could argue that the term EMCD with C standing for circular or chiral is still justified in this setup. Indeed, the in-plane components will lead to a phase signature of the inelastic electrons that will resemble a pi-orbital rather than the typical vortex state that is involved with an out of plane magnetic excitation. The term EMLD with L for linear might be more appropriate.

Answer:

We thank the referee for this question, which goes deep into principles behind the phenomenon. We believe that 'C' as circular/chiral is the correct notion in this case. The terminology EMCD/EMLD comes from analogies with x-ray based techniques. X-ray magnetic circular dichroism is an effect that is proportional to magnetization (e.g., it is zero for antiferromagnets), while x-ray magnetic linear dichroism is an effect that is proportional to square of magnetization (thus nonzero for antiferromagnets). They have somewhat different physical origins as well. For instance, the linear dichroism strongly depends on exchange splitting of core states, while circular dichroism is only very weakly sensitive to the exchange splitting of core states. In our simulations, the magnetic signal comes from a term $(\mathbf{q} \times \mathbf{q}') \cdot \mathbf{M}$ and for electrons $(\mathbf{q} \times \mathbf{q}')$ can have a direction other than the direction of propagation

(this is in contrast to x-rays, which are only sensitive to a projection of magnetization onto propagation direction). The effect is still directly proportional to magnetization (not M^2) and the exchange splitting of core states was not even considered in our simulations. Therefore, in our opinion the EMCD is still the appropriate acronym/nomenclature for describing this method of detection of in-plane magnetization.

(2) an estimate of the sensitivity of this method with respect to strain would be important to verify if small amounts of strain could not lead to spurious measurement. Moreover, this strain could couple to the magnetic state, making the interpretation even more convoluted.

Answer:

Small amounts of strain will lead to a slight distortion of the crystal structure and may also further change the magnetic state for materials. However, the change in crystal structure due to strain is very small (usually at most several percent). Therefore, the relative intensity of the EMCD signal will not change significantly, unless the magnetic state (e.g. magnetic moment) changes too much. To address this possibility, we perform a simulation for *hcp* Co with both -3% compressive strain and 3% tensile strain in the (001) plane, which leads to a change in lattice parameters. To simplify this, we assume that both the lattice parameter *a* and *b* are changed by -3% or 3%, respectively. The lattice parameter *c* is calculated by assuming that the volume of the unit cell is not changed. The results are as follows, and we find that there is almost negligible difference between these different strain states. It should be noted that the magnetic moment is assumed to be same for all these three cases here. However, if the strain will induce a large change in the magnetic moment, then it may be possible to detect an obvious change in EMCD signal in the experiments. We have added this point in our Supplementary Information.

Figure. Effect of strain on the relative intensity of the EMCD signals for *hcp* Co under the (-210) three-beam case with the thickness of 20 nm and the acceleration voltage of 300 kV. The upper panel, middle panel and lower panel correspond to the strain of 3%, 0% and -3% in the (001) plane, respectively.

Revision:

In our revised Supplementary Information, we have added the following part.

Effect of strain on the intensity of the EMCD signal

Small amounts of strain will lead to a slight distortion of the crystal structure and may also further change the magnetic state of the materials. Here, we perform a simulation for *hcp* Co with both -3% compressive strain and 3% tensile strain in the (001) plane, which will lead to a change in lattice parameters. To simplify this, we assume that both the lattice parameter *a* and *b* are changed by -3% or 3%, respectively. The lattice parameter *c* is calculated by assuming that the volume of the unit cell is not changed. The results are shown in Fig. S8. We find that there is almost negligible difference in the relative intensity of the EMCD signal between the different strain states. It should be noted that the magnetic state is assumed to be same for these three cases. However, if the strain will result in a significant change in the magnetic moment, then it may be possible to detect the obvious change in EMCD signal experimentally.

Figure S8. Effect of strain on the relative intensity of the EMCD signals for *hcp* Co under the (-210) three-beam case with the thickness of 20 nm and the acceleration voltage of 300 kV. The upper panel, middle panel and lower panel correspond to the strain of 3%, 0% and -3% in the (001) plane, respectively.

(3) an evaluation of the total sampled amount of material versus signal to noise ratio could provide an estimate of the application limits of this technique and of the fundamental limits avoiding this method to go to atomic resolution mapping.

Answer:

After we submitted this manuscript, we became aware of a new published paper PHYSICAL REVIEW B 94, 134430 (2016), which provides a detailed and systematic SNR analysis to quantify the confidence of EMCD signal detection at high spatial resolution. To do this evaluation, a large amount of EMCD spectra are acquired under the STEM mode, which allows to obtain EMCD signals with different intensities and SNR by the integration of different lengths of lateral scanned areas. For our present

in-plane EMCD technique which works in Lorentz mode, a Lorentz-STEM mode with the objective lens off is needed to achieve such a database. Unfortunately, this capability was not available on the microscope that we used. There might also be other unexpected problems during the experiments as a result of the critical demand from the EMCD diffraction geometry, especially in STEM mode. Besides, in STEM mode the in-plane EMCD technique will become more complicated than for the out-of-plane case. The fact that the in-plane EMCD signals are localized on the systematic reflection axis, in which the diffracted discs are strongly overlapped, may also have a large impact on the SNR. These problems will be assessed in the future using in-plane Lorentz-STEM-EMCD experiments, and theoretical simulations performed for a converged beam rather than a parallel beam here.

However, we give an estimate of noise by evaluating the noise fluctuations in the pre-edge edge of EMCD spectra. The noise is for Figs 4(a), 4(b) and 4(c) are $\sim 0.27\%$, $\sim 0.28\%$ and $\sim 0.30\%$ (it is relative to the peak intensity of spectra at L_3 edge), respectively. The noise for Figs 5(a) and 5(b) are $\sim 0.34\%$ and $\sim 0.34\%$, respectively. This information has been added to the revised manuscript.

Revision:

We have added the follow sentences to the “Discussion” part of our revised manuscript.

The present experiments are conducted in Lorentz-TEM mode at medium resolution. The Lorentz-STEM mode could be used in the future to achieve much higher spatial resolution. An assessment of the sampled amount of material versus signal to noise ratio, as reported in ref [30], could then be used to provide an estimate of the resolution limit. Theoretical analysis should then also be performed to discuss the impact of overlapped diffracted discs on the in-plane EMCD signals, which are just localized on the systematic reflection axis.

[30] Thersleff, T., Rusz, J., Hjärvarsson, B. & Leifer, K. Detection of magnetic circular dichroism with subnanometer convergent electron beams. *Physical Review B*, 94(13): 134430 (2016).

(4) the paragraph from line 123 to 131 is correct but could be explained in a clearer way.

Revision:

We have rewritten this paragraph as follows,

The EMCD signal also differs in intensity between the three components, as it depends on the relative magnitudes of the x , y and z components of momentum transfer. The q_y component originates primarily from the placement of the detector relative to the transmitted beam. For q_x , not only the contribution from the detector placement, but also the strength of the excited Bragg spots contributes additional momentum transfer to q_x in the systematic row direction. For q_z , the primary origin is the energy loss of several hundreds of eV, whose contribution is rather small. The products $\mathbf{q} \times \mathbf{q}'$ that contain q_x are therefore expected to produce the strongest signals, as shown in Figs 2(c) and 2(d) for the y and z directions. In contrast, the x component of the EMCD signal will be weak, as shown in Fig. 2(b), as only q_y and q_z are relevant.

(5) shifting the diffraction geometry during the experiment is in practice cumbersome. Are there other ways to selectively measure x and y component of the magnetization separately?

Answer:

Under the three-beam diffraction geometry, there are no other ways to measure the x and y component of the magnetization simultaneously. Shifting the diffraction geometry might be the only choice until now. However, the zone axis diffraction geometry for a crystallographic direction with very high symmetry may help solve this problem and allow the three magnetic components to be extracted simultaneously from a single diffraction pattern, as we state in the “**Discussion**” part of our original manuscript, such as for bcc Fe under [001] zone axis in ref [10].

[10] Ruzs, J., Rubino, S., Eriksson, O., Oppeneer, P. M. & Leifer, K. Local electronic structure information contained in energy-filtered diffraction patterns. Phys. Rev. B 84, 064444 (2011)

(6) is there a reason why the authors didn't perform an energy filtered diffraction experiment revealing the full details of the scattering to be compared to the theory. If I understand correctly the experiment only verifies a few fixed positions in this pattern which still could mask severe differences between experiment and theory.

Answer:

It was also our initial intention to further perform an energy filtered diffraction pattern experiments, as they can provide the distribution of EMCD signals in the whole diffraction pattern and would be directly comparable to theoretical simulations, as the referee states. However, there are several reasons that prevented us from successfully conducting such experiments.

In Lorentz mode, the camera length becomes very large because the objective lens is turned off. Therefore, we cannot observe the whole effective diffraction pattern on the CCD camera in order to accurately locate the detector positions for signal acquisition. There are two approaches that can be used to solve this problem now.

First, we can perform experiments on the Titan PICO using a Flu Camera to record the diffraction pattern on the fluorescent screen. The position of the EELS entrance aperture can also be located in the image. Then, we can use the large entrance aperture to choose the detector positions for signal acquisition.

Second, we can switch to the energy-filtered Lorentz-TEM mode to reduce the camera length. However, energy-filtered Lorentz-TEM mode requires a good alignment between the Lorentz mode and energy-filtered image/diffraction mode. Unfortunately, this mode was not well aligned and we can not obtain the required image of a filtered diffraction pattern. Moreover, the energy-filtered Lorentz-TEM mode introduces additional distortions in the diffraction pattern. We hope that these problems will be well addressed in the future.

(7) line 305 'home-written' seems to imply that due to lack of time at the lab, the software needed to be written from home.

Revision:

The “home-written” has been removed from our revised manuscript.

Reviewer #1 (Remarks to the Author):

[The referee believes the manuscript can be published, with no further comments for the authors]

Reviewer #2 (Remarks to the Author):

The issues were adequately addressed by the authors and the paper is now in acceptable form.